# *GDAP1* Is Dysregulated at DNA Methylation and H3K4me3 Levels in Alcohol Use Disorder

**DOI:** 10.3390/ijms26041623

**Published:** 2025-02-14

**Authors:** Emilia Kawecka, Henning Plättner, Lena Ederer, Kilian Niemann, Sarah Pasche, Milan Zimmermann, Susanne Edelmann, Vanessa Nieratschker

**Affiliations:** 1Department of Psychiatry and Psychotherapy, University Hospital of Tuebingen, Eberhard Karls University of Tuebingen, 72076 Tuebingen, Germany; emilia.kawecka@tum.de (E.K.); vanessa.nieratschker@med.uni-tuebingen.de (V.N.); 2German Center for Mental Health (DZPG), Partner Site Tuebingen, 72076 Tuebingen, Germany

**Keywords:** *GDAP1*, alcohol use disorder, H3K4me3, DNA-methylation, epigenetics, gene expression

## Abstract

Alcohol use disorder (AUD) contributes significantly to the global burden of disease. The susceptibility for AUD is mediated through an interaction of genetic risk factors and environmental influences. These gene × environment (G × E) interactions manifest as epigenetic regulations of gene expression, among other things. Previous research suggests an association between Ganglioside Induced Differentiation Associated Protein 1 (*GDAP1*) DNA methylation and AUD. Here, we investigate the epigenetic dysregulation of *GDAP1* in AUD through comparing DNA methylation in whole blood and saliva, as well as H3K4-trimethylation (H3K4me3) in PBMC (Peripheral Blood Mononuclear Cell) samples of AUD patients and healthy control individuals. Additionally, the effect of abstinence-based therapy was investigated. AUD patients before treatment exhibit significantly lower promoter DNA methylation levels in whole blood and saliva, as well as lower H3K4me3 near the transcription start site. *GDAP1* gene expression was not significantly altered. Following treatment, H3K4me3 was significantly increased in patients and no longer differed from control individuals. There was no significant effect of treatment on DNA methylation. We conclude that *GDAP1* is epigenetically dysregulated in AUD patients, and is responsive to abstinence-based therapy at the level of H3K4me3. It should be investigated further to establish its potential as a diagnostic biomarker.

## 1. Introduction

Alcohol is an addictive and psychoactive substance with major health implications. It has been ranked as the third most relevant health risk factor next to high blood pressure and smoking according to a global burden of disease study in 2010 [1]. Despite its risks, alcohol consumption is socially accepted. Industrialized countries commonly exceed the global average per capita consumption of 5.5 L, especially in Europe which averages 9.2 L. As of 2019, Germany is categorized as a high consumption country with an average of 12.2 L pure alcohol consumed annually per person [2,3].

Alcohol use disorder (AUD) is characterized by the harmful use of alcohol and the associated negative health outcomes of both psychological and physiological nature. Patients experience a lack of control towards alcohol consumption, strong cravings, buildup of tolerance over time, and withdrawal symptoms when abstaining from consumption [4].

Genetic predispositions, such as a family history of AUD, increase the probability of occurrence [5,6], and even adopted-away children display a higher risk of AUD if their biological parents suffered from the disease [7]. Twin studies suggest a heritability of 50 to 60%, with higher numbers being found in monozygotic twins [8,9]. However, AUD risk factors are not limited to genetic variants, as the environment also plays a critical role in its development. Monozygotic twins that spent longer times living together have higher chances of co-AUD diagnosis [10], which could be explained by an interplay between genetic factors and the environment. These interactions are referred to as gene × environment interactions (G × E) [11] and are mediated through epigenetic regulation among other things. Epigenetic mechanisms alter the regulation of genes through different modifications without changing the DNA sequences themselves [6,11]. These modifications warrant the time- and context-specific suppression or activation of genes through, e.g., altered DNA accessibility [12] and respond to environmental factors such as exposure to alcohol [11,13,14]. DNA methylation (DNAm) is an epigenetic modification based on the addition of a methyl group to cytosines by DNA-methyltransferases [15]. It most commonly occurs at cytosines directly followed by a guanine, a so called CpG site (CpGs). Regulatory regions of the gene, including the promoter, are frequently marked by an abundance of CpGs, forming CpG islands [16]. DNAm in those regulatory regions is generally negatively associated with gene expression [17].

DNA in the nucleus is organized as chromatin, which is a complex of packaging proteins—approximately 50% of which are histones—and DNA. In addition to DNAm, modifications of the amino-terminal histone tails play a highly complex role in regulating gene expression. Depending on the type (e.g., methylation), the location, and valence of these modifications, as well as the interaction of different histone alterations, they can activate or repress gene expression [18]. An important epigenetic mark is H3K4me3, the trimethylation of the 4th lysine in histone 3 of the nucleosome. It mostly occurs near transcription start sites (TSS) of genes and is linked to transcriptionally active chromatin and, therefore, often referred to as an active histone mark [19,20].

Over the past few decades, more and more insight into the epigenetic dysregulation in AUD has been gained. Epigenome wide association studies (EWAS) of DNAm discovered many differentially methylated genes in patients suffering from AUD, of which 184 could be replicated in at least two independent studies [21]. Differential DNAm in *GDAP1* has been identified in one of the first EWAS on AUD [22], and was later replicated by our group [23]. *GDAP1* encodes for Ganglioside Induced Differentiation Associated Protein 1. It is primarily expressed in neuronal cells, especially in the brain [24]. The protein functions as an atypical glutathione-S-transferase which catalyzes the bond formation between glutathione and toxic metabolites for subsequent excretion. This pathway plays a significant role in the detoxification and the redox balance of the cell. Knockout experiments showed increased oxidative stress in cells, coupled with diminished glutathione concentration [25,26]. Defects in the gene are associated with Charcot–Marie–Tooth disease, a neurodegenerative disease that affects the peripheral nervous system and causes disrupted motor function [27,28,29].

In our own study investigating DNAm levels in the promoter region of *GDAP1*, we were able to replicate the hypomethylation in patients suffering from AUD previously described by Philibert et al. [22] Furthermore, we showed that this hypomethylation also correlated with AUD severity and reverted to levels observed in healthy control individuals after three weeks of detoxification therapy and abstinence [23].

The present study aimed to examine the epigenetic dysregulation of *GDAP1* in the context of AUD in more detail by investigating DNAm of the promoter region in whole blood and saliva, as well as H3K4me3 in PBMCs (Peripheral Blood Mononuclear Cells). In addition, the impact of those modifications on gene expression levels have been analyzed and the effects of a three-week inpatient detoxification treatment have been evaluated, with the goal to determine the potential of *GDAP1* as a diagnostic epigenetic marker of AUD.

## 2. Results

The study sample included 74 patients and 86 healthy control individuals (Appendix A). A schematic overview of our study design is displayed below (Figure 1). While age, sex, and smoking behavior of both groups were tried to be matched throughout the recruitment process, the two groups still revealed some significant differences: Control individuals were, in general, about a decade younger and included more non-smokers and females than patients (Table 1). Not all epigenetic markers were analyzed in all participants, therefore yielding lower sample sizes in the single analysis steps.

Alcohol Use Disorder Identification Test (AUDIT) scores were significantly higher in patients at both timepoints (Table 1, *p* < 0.001). Obsessive-compulsive Drinking Scale (OCDS) scores were also higher at both timepoints (Table 1, *p* < 0.001), which shows elevated craving and obsessive tendencies towards alcohol in AUD patients. Patients also displayed higher Global Severity Index (GSI) scores at both timepoints (Table 1, *p* < 0.001), exceeding the psychological distress threshold of 60 at T1. All questionnaire scores improved post therapy (Appendix A), showing that the detoxification had positive effects on drinking behavior and psychological welfare of patients.

### 2.1. Initial Analysis Revealed Altered Molecular Markers in AUD Patients, Independent of Sex, Age, or Smoking Behavior

A linear mixed model approach was applied to our studied epigenetic markers, namely whole blood and saliva *GDAP1* DNAm, as well as H3K4me3 in the *GDAP1* promoter. Sex, age, and smoking behavior were included in the analysis, in addition to a slope for random effects caused by interpersonal differences and repeated measures. To ensure the proper model fit, all data were log-transformed beforehand in order to fulfill the model’s requirement of normal distribution.

Whole blood *GDAP1* DNAm was neither associated with age or smoking behavior of the participant, nor with timepoint. As expected [22,23], *GDAP1* DNAm in whole blood was significantly altered in patients compared to healthy control individuals (*p* < 0.001, Table 2). Additionally, sex had a significant effect on *GDAP1* DNAm (*p* = 0.05, Table 2). A low residual standard deviation of 0.1297 in DNAm levels suggests that most variance in methylation levels can be explained by the included fixed and random effects.

Saliva DNAm of *GDAP1* was neither associated with sex nor smoking, but it was significantly associated with age (*p* = 0.003, Table 2) as well as with timepoint (*p* = 0.01, Table 2). Methylation was also altered in patients in comparison to healthy control individuals (*p* < 0.001, Table 2). Residual standard deviation was quite low (0.1731, Table 2).

*GDAP1* H3K4me3 Fold change was not dependent on age, sex, timepoint, and smoking behavior of the participants. Once again, patients had significantly altered values compared to healthy control individuals (*p* = 0.015, Table 2). The residual standard deviation was high (1.315, Table 2), suggesting that other, unconsidered factors, might contribute to the differences in values between participants.

Though group distributions varied strongly (Table 1), this cofounding analysis revealed that major effects on the epigenetic marks are driven by group (AUD patients vs. healthy control individuals). Subsequently, additional post hoc tests were applied to determine specific differences between patients and healthy control individuals.

### 2.2. Lower DNAm Levels in GDAP1 Promoter in Whole Blood Samples of AUD Patients Compared to Healthy Control Individuals

The *GDAP1* DNAm level was significantly lower in AUD patients in comparison to healthy control individuals in whole blood samples. Three different CpG sites, which were previously found to be differentially methylated by Brückmann et al. [23], were analyzed. Due to their proximity and significant strong correlation (Appendix A), the methylation levels were averaged. Healthy control individuals revealed an average *GDAP1* DNAm level of 6.49 (±1.08) % (T1) and 6.67 (±1.22) % (T2), while AUD patients displayed lower *GDAP1* DNAm levels at both timepoints, with 4.86 (±1.20) % at T1 and 5.12 (±1.15) % at T2. The difference between patients and healthy control individuals was highly significant at both timepoints (*p* < 0.001, Figure 2).

### 2.3. GDAP1 Promoters Display Lower DNAm Levels in Saliva of AUD Patients Compared to Healthy Control Individuals

Saliva *GDAP1* methylation, which was not investigated so far, is also altered in AUD patients compared to healthy control individuals. The same CpG-sites were tested in saliva samples and averaged for further analysis (Appendix A). Similar to the whole blood samples, a lower average *GDAP1* DNAm was observed when comparing patients to healthy control individuals, with 2.90 (±0.81)% and 3.57 (±0.80)% (T1), as well as 2.64 (±0.57)% and 3.17 (±0.58)% (T2), for the respective groups. The *GDAP1* methylation levels differed significantly between patients and healthy control individuals (*p* < 0.001, Figure 3) at both timepoints. This suggests that AUD-mediated effects on DNAm of *GDAP1* are not limited to blood and can also be measured in alternative peripheral tissues.

### 2.4. Reduced H3K4me3 Levels near TSS of AUD Patient PBMCs Compared to Healthy Control Individuals

The histone modification H3K4me3 near the TSS of *GDAP1* is less abundant in AUD patients at T1. Healthy control individuals displayed an average fold change of 2.78 (±1.74) at T1, which was significantly higher compared to the fold change measured in patients (2.02 (±2.19), *p* = 0.015, Figure 4). At T2, patients and healthy control individuals revealed a similar H3K4me3 abundance, with control individuals having an average fold change of 2.39 (±1.28) and patients 2.25 (±2.00, *p* = 0.285). These results serve as an indicator for epigenetic dysregulation of *GDAP1* in AUD beyond DNAm levels.

### 2.5. Effects of AUD Therapy on H3K4me3 as Well as DNAm of Whole Blood and Saliva in GDAP1

The H3K4me3 fold change increased significantly from T1 to T2 and did not differ from control values anymore (Figure 4). This suggests that the 3-week treatment influenced fold change in the histone modification at the *GDAP1* promoter in a positive way, reverting its values to control-comparable levels. *GDAP1* DNAm did not show this reversion in both blood and saliva of AUD patients.

### 2.6. No Difference in Gene Expression Levels of GDAP1 in Whole Blood of AUD Patients Compared to Healthy Control Individuals

To determine whether the epigenetic dysregulation of *GDAP1* affects gene transcription, a *GDAP1* gene expression analysis was performed. No significant difference between patients and healthy controls was detected, neither at T1 (Patients: 0.94 log2FC, healthy control individuals: 1.20 (*p* = 0.21, Figure 5)), nor at T2 (Patients: 0.91 log2FC, healthy control individuals: 1.18 (*p* = 0.21, Figure 5)).

## 3. Discussion

This study shows a DNA promoter hypomethylation of *GDAP1* in whole blood samples of AUD patients compared to healthy control individuals and therefore successfully replicates previous results [22,23]. Additionally, it describes two newly identified, AUD-related dysregulated epigenetic marks in *GDAP1*, namely promoter DNA hypomethylation in saliva and reduced H3K4me3 at TSS close regions in PBMCs. Furthermore, H3K4me3 is upregulated after a 3-week inpatient detoxification therapy and no longer significantly different from healthy control individuals.

Blood DNAm results are in concordance with Brückmann et al., who found whole blood *GDAP1* DNAm levels in patients to be 1.2% lower in comparison to control participants [23]. Our finding underlines the previously established dysregulation of blood methylation patterns in *GDAP1* in AUD patients. In addition, we show that this dysregulation is also detectable in saliva samples. This could be of value for the establishment of *GDAP1* promoter DNAm as a diagnostic biomarker, as saliva represents a non-invasive tissue which is easier to obtain and handle in clinical, but especially in ambulatory settings.

Detoxification therapy did not have statistically significant effects on DNAm of *GDAP1* in blood and saliva, although the mean difference in levels between patients and healthy control individuals at T2 was reduced in both tissues (Figure 1 and Figure 2). Detoxification treatment might not have an effect on the epigenetic regulation of *GDAP1* after a relatively short period of time. Since the significant increase in DNAm in whole blood at T2, as previously described by Brückmann [23], could not be replicated in our study, further work in larger cohorts is needed to verify the effects of therapy on DNAm levels of *GDAP1*.

In addition, we report reduced PBMC H3K4me3 in a region near the TSS of *GDAP1* in AUD patients, which reverts back to the level of control individuals following detoxification treatment. Although these epigenetic dysregulations were found in different tissues and would need to be compared in the same material in the future to establish a possible correlation between DNAm and H3K4me3. DNA hypomethylation combined with decreased H3K4me3 is a rather unusual finding. Most studies indicate an inverse relationship between these two epigenetic marks [31,32,33]. Generally, reduced promoter DNAm is associated with higher H3K4me3 levels [31], and therefore leads to an increase in gene expression [34]. However, the finding of reduced H3K4me3 in combination with a seemingly, although not statistically significant, lower gene expression, hints towards a complex epigenetic dysregulation of *GDAP1* in AUD patients. H3K4me3 is highly relevant for the activation of gene transcription, possibly playing an instructive role in enhancing the chromatin accessibility for transcription factors [35,36]. It is found near the TSS of nearly 75% actively transcribed genes [37]. However, other histone marks, like H3K9me3 and H3K27me3, are associated with repression of gene expression [38,39]. Depending on the presence of these opposing marks and their distribution in comparison to H3K4me3, gene expression can be altered in different ways [40]. Additionally, DNAm has been found to not necessarily lead to transcriptional repression, and can also co-exist with activating histone marks, such as H3K4me3, leading to transcriptionally accessible chromatin [41]. These studies highlight the potentially over-simplified and case-dependent current understanding of the role of single epigenetic marks in the regulation of gene expression. To obtain a clearer picture of how exactly *GDAP1* is dysregulated in AUD patients, it is of utmost importance to study repressive histone marks and their influence, as well as repeat the gene expression analysis with a larger sample size.

The differences in *GDAP1* DNAm in blood and saliva, as well as H3K4me3, between patients and controls are significant, albeit comparatively small. Identifying AUD while only relying on epigenetic *GDAP1* dysregulations would therefore be challenging in individual patients, since the differences may prove to be miniscule. However, *GDAP1* DNAm and H3K4me3 could be used in diagnostic marker panels with other AUD-related alterations to improve diagnostic strength. For example, High Mobility Group Box Protein 1 (*HMGB1)* expression was found to be elevated in individuals with binge-drinking tendencies [42]. *HECW2* (HECT, C2, and WW Domain Containing E3 Ubiquitin Protein Ligase 2) was also found to be differentially methylated in AUD patients before detoxification therapy [43]. Testing these alterations in epigenetic marks and expression patterns in AUD patients in a blood-based panel could be a promising option for diagnostic measures.

It is also important to note that DNAm and gene expression were measured in whole blood and saliva, whereas H3K4me3 was analyzed in PBMCs. Results may therefore be dependent on tissue-specific cell type composition in samples. While suboptimal for compatibility, the utilization of PBMCs for examining the epigenetic dysregulation of histones was integral as to the best of our knowledge, there is no protocol available for the analysis of histone modifications using whole blood. The limitation of comparing *GDAP1* DNAm and gene expression in whole blood to H3K4me3 in PBMCs makes it harder to form causal relationships between the two epigenetic features and gene expression. PBMCs could display alternative epigenetic marks compared to whole blood samples, which contain PBMCs, but also polymorphonuclear leukocytes. Therefore, it is of utmost necessity to assess *GDAP1* DNAm and gene expression in PBMCs in future studies to confidently say that the aberrant pattern of H3K4me3 occupancy and DNAm in *GDAP1* combined with seemingly, although not statistically significant, lower gene expression is present in AUD patients and is not attributable to tissue specific differences. PBMC isolation adds an additional work step into the analysis of epigenetic patterns, which would complicate clinical applications, but would ensure a more detailed picture of all implicated epigenetic marks.

Philibert determined the altered CpG methylation of *GDAP1* in PBMCs [22]. A standardized analysis of epigenetic markers would improve comparability and make the results more robust. However, PBMC isolation is also a logistically challenging work step that would complicate the use of *GDAP1* DNAm as a biomarker in future clinical settings.

It is also worth noting that the DNAm analysis in blood and saliva was limited to three CpG sites in close proximity due to the methodology used. Pyrosequencing yields excellent, reliable analyses of DNAm patterns, but only in shorter sequences not exceeding 60 base pairs. Analyzing the whole CpG island, or even the entire *GDAP1* gene, would, therefore, not have been possible in our study. Choosing pyrosequencing ensured a proper reflection of DNAm patterns but came at the price of limited spatial analysis. Future studies should cover the entire CpG island through the use of different sequencing methods. Unexpectedly, the investigated variables are not confounded by the smoking behavior of the participants, although the effects of smoking and alcohol are often tightly linked to one another [44]. For instance, a multitude of risk genes associated with AUD are also risk factors for nicotine addiction [44,45,46]. A past study in mice also implicated nicotine-dependent reduced *GDAP1* expression in hippocampal tissue [47]. Our results put the use of *GDAP1* as a general addiction marker into question. However, it is important to mention that smoking behavior was not equally distributed amongst participants, as most patients exhibited comorbid tobacco use disorder, and only a small subset of healthy control individuals revealed the same behavior. In most cases, DNAm was not confounded by sex and age, which is in concordance to prior works [23,43]. This would implicate *GDAP1* as a robust diagnostic marker for AUD, independently from patients’ demographic features. However, the confounding effect of sex on blood DNA methylation, as well as age on saliva DNA methylation levels need to be explored further and with matched cohorts, as it may also stem from the demographic differences between patients and control individuals.

*GDAP1* expression tends to be lower in AUD patients compared to healthy control individuals, and although not statistically significant, it supports the hypothesis of H3K4me3 being an important transcriptional activator that potentially co-exists with and overrules DNA methylation patterns [41,48]. Reduced *GDAP1* expression has been found to increase cellular oxidative stress and cause problems in mitochondrial network formation [25]. When oxidative stressors, such as reactive oxygen species (ROS), vastly exceed the cellular antioxidant capacities, the cell can induce death through apoptosis or necrosis [49]. As *GDAP1* is mostly expressed in neuronal tissues [50] lower *GDAP1* expression may lead to increased oxidative stress in AUD patients, which causes peripheral nerve damage and general neural degeneration [51]. About 50% of all chronic alcohol consuming individuals experience alcohol-related peripheral neuropathy (ALN), a neurodegenerative disease warranting disruptions in sensory and motor function [52,53]. Initial symptoms include burning sensations throughout the patient’s body and altered receptiveness or reactivity to pain. Established ALN causes weakness in extremities, beginning in the hands and spreading towards the core body [52,54], similarly to GDAP1-deficiency related Charcot–Marie–Tooth disease [27,28,29]. ALN belongs to the group of peripheral neuropathies, which have been associated with an increase in ROS and a decrease in intracellular antioxidants [55,56,57]. This constitutes the death of peripheral nerves [56,58]. It is therefore possible that increased oxidative stress in AUD patients due to epigenetically dysregulated *GDAP1* expression supports the deterioration of peripheral nerves and the progression of ALN. ALN severity is correlated with the duration and extent of alcohol intake throughout life, as it is more common in frequent and heavy drinkers than episodic ones [52,59]. *GDAP1* expression could potentially correlate with both time since AUD diagnosis and heavy drinking and, additionally, be lower in patients with established ALN. Brückmann et al. suggested a negative relationship between AUD severity and DNAm [23], a finding we were able to replicate in our cohort (Appendix A). Future work could decipher whether *GDAP1* expression is lower in heavy, long-term drinkers and ALN patients.

This work indicates that the epigenetic dysregulation of *GDAP1* in AUD patients reaches beyond blood DNAm levels and implies saliva DNAm and H3K4me3 as additionally altered marks. We therefore successfully validated CpG-site specific *GDAP1* DNA hypomethylation in whole blood, as implicated in past studies [22,23], while extending the findings to reduced *GDAP1* DNAm in saliva, lower *GDAP1* H3K4me3 in PBMCs, and a seemingly lower (although not statistically significant) gene expression. Our study provides further evidence for an epigenetic dysregulation of *GDAP1* in AUD patients not only at the level of DNAm, but also on histone modification levels. The association between AUD and the epigenetic regulation of *GDAP1* on multiple levels implicates its potential use as a biomarker for AUD that needs to be validated further in studies using the same tissue to analyze DNAm and histone modifications, preferably PBMCs.

## 4. Materials and Methods

### 4.1. Study Sample

Blood samples were collected at the Department of Psychiatry and Psychotherapy, University Hospital, Tuebingen, Germany. 74 patients with AUD (26 female, 48 male) and 86 healthy control individuals (55 female, 31 male) participated in the study. All participants were of European descent. The AUD diagnosis was based on the criteria of the International Classification of Diseases, 10th revision (ICD-10 [4]). AUD patients were subjected to a 3-week inpatient treatment program, where they went through alcohol withdrawal and therapy. Blood and saliva samples as well as questionnaires were collected before (T1) and after the 3-week therapy (T2). Samples and data of control individuals have also been collected twice, with an interval between T1 and T2 of three weeks. Smoking behavior was assessed through the number of cigarettes consumed daily. The AUDIT was administered at both timepoints to assess drinking frequency and related behavioral patterns. The OCDS measured alcohol craving. The Symptom Checklist revised (SCL-90^®^-R) was utilized to measure psychological and physiological distress. Individual scores were converted to the age and sex specific GSI, which rates the subjective burden of a patient. A GSI over 60 is indicative for psychological distress. The study was approved by the ethics committee of the University of Tuebingen (Reference number 264/2018 BO2) and was conducted in accordance with the Declaration of Helsinki.

### 4.2. DNA-Methylation Analysis in Whole Blood and Saliva

Ethylenediaminetetraacetic tubes (EDTA, BD, Heidelberg, Germany) and saliva samples (Oragene^®^ DNA Collection Kits, DNA Genotek, Ottawa, ON, Canada) were collected at both timepoints (T1 and T2). The DNA was extracted from blood samples using the QIAamp^®^ DNA Blood-Maxi Kit (Qiagen, Hilden, Germany [60,61]), and with Oragene^®^ prepIT•L2P for saliva samples (DNA Genotek, Ottawa, ON, Canada), respectively, according to the manufacturer’s instructions. 500 ng genomic DNA was bisulfite converted using the EpiTect Fast Bisulfite Kit (Qiagen) [62,63] according to the manufacturer’s instructions. Bisulfite converted DNA was eluted in 20 µL elution buffer and stored at −20 °C until further analyses.

A PCR was performed to amplify the promoter region of *GDAP1*. Primer sequences were as follows: forward primer: 5′-ATT TTT AGG TTT GTT AGG GGT TTT TTA GT-3′; reverse primer: 5′-Biotin-ACT TCT CCC TCC CAC ACT ACC C-3′). All PCRs were performed following the manufacturer’s protocol. The *GDAP1* CpG Assay included 3 CpG sites upstream of the *GDAP1* TSS within chromosome 8 (site 1 located at 74,350,287; site 2 located at 74,350,296; and site 3 located at 74,350,298, GRCh38/hg38). CpG site 1 corresponds to cg23779890, the site which has been previously implicated by Brückmann et al. and Philibert et al. [22,23]. Specificity and quality of the PCR products was checked on a 2% agarose gel. DNAm levels were analyzed by pyrosequencing using the PyroMark Q24 system with the sequencing primer (seq): 5′-GTT TGT TAG GGG TTT TTT A-3′ and the corresponding PyroMark Q24 Software 2.0 (Qiagen) [64,65]. Each sample was amplified twice and both amplicons sequenced as technical replicates. The mean percentage was used for further analyses. However, replicates revealing a deviation ≥3% were repeated. To detect disparate amplification of unmethylated DNA fragments, a titration assay using standardized bisulfite-converted control DNA samples (EpiTect Control DNA, Qiagen) with established DNA methylation levels of 0%, 25%, 50%, 75%, and 100% was performed.

### 4.3. H3K4-Trimethylation Analysis in Peripheral Blood Mononuclear Cells (PBMCs)

To determine the H3K4me3 fold change in *GDAP1*, a protocol including chromatin immunoprecipitation (ChIP) and subsequent quantitative polymerase chain reaction (qPCR) was conducted. Blood was drawn in BD Vacutainer^®^ CPT™ tubes (BD Bioscience, Franklin Lakers, NJ, USA) and processed according to the manufacturer’s protocol. The DNA and proteins were crosslinked using 37% formaldehyde and the reaction was stopped with 1.25 M glycine. The samples were purified by washing twice with 1X Phosphate-Buffered Saline (PBS) including centrifugation steps at 3000× *g* at room temperature for 5 min. RIPA SDS buffer containing protease inhibitors was added to induce cell lysis. Chromatin fragmentation was achieved through sonication with the Bioruptor^®^ pico sonicator (Diagenode Inc., Denville, NJ, USA) by using 16 cycles consisting of 30 s shearing and 30 s resting period. The immunoprecipitated sample (IP) was directly stored at −80 °C. The input control (IC) was incubated overnight at 65 °C in a water bath and purified using the QIAquick PCR Purification Kit (Qiagen, Hilden, Germany). The IC was used as a positive control in order to differentiate between non-specific background signals in the genome from H3K4me3-specific signals in the first exon region of GDAP1, so that the results could be put into perspective. A gel electrophoresis was performed as a sonication control. Samples were immunoprecipitated with a H3K4me3 recombinant rabbit monoclonal antibody (Thermo Fisher Scientific, Waltham, MA, USA, ID: 9H28L57), which was coupled with magnetic beads and then incubated with the samples overnight at 4 °C. Additionally, IgG-rabbit antibody was run as a negative control to ensure minimal non-specific binding (Cell Signaling Technology, Inc., Danvers, MA, USA). The immuno-bound DNA was purified using the QIAquick PCR Purification Kit (Qiagen, Hilden, Germany).

The first exon region of *GDAP1* including the TSS (chr8:74,350,353–74,350,750, GRCh38/hg38) was amplified using the following primers: (forward: 5′ ACGCGGAGGTTAAGCTCATT 3′, reverse: 5′ GACACTGGAGGCGGATTTCT 3′. *GAPDH*, a housekeeping gene with steady activity, was used as positive control and amplified (chr12:6,534,651–6,535,051) using the following primers: forward: 5′ TCA GAC ACC ATG GGG AAG GT 3′, reverse: 5′ CTT GAG CTC TCC TTG CGGG 3′. A negative control was used to rule out any potential DNA contamination. The EvaGreen MasterMix (HOT FIREPol*R* EvaGreen*R* qPCR Mix Plus, Solis BioDyne OU, Tartu, Estland 08-24-0000S) was used. Samples were run in triplicates for both IC and IP. Cycle threshold (CT) values exceeding a threshold of 35 signified poor sample quality and were excluded from further analysis. The QuantStudio PCR run raw data were used for fold (F) enrichment calculation. Fold enrichment indicates how strongly H3K4me3 is enriched in *GDAP1* compared to the housekeeping gene *GAPDH*, relative to the general abundance of DNA determined in the IC. The average CT value for each triplicate was calculated and put into proportion to the IC. Afterwards, the values were placed in relation to the housekeeping gene *GAPDH* and that value was potentiated in order to obtain the fold enrichment. A fold enrichment exceeding 1 suggests that H3K4me3 is enriched in *GDAP1*.

### 4.4. Gene Expression

Peripheral whole blood samples were collected in PAXgene Blood RNA Tubes (IVD) according to the instructions provided by the manufacturer (PreAnalytiX GmbH; Hombrechtikon, Switzerland), and subsequently stored at −80 °C until further use. The frozen samples were thawed and incubated at room temperature for 2 h before proceeding to RNA extraction. Total RNA was extracted using the PAXgene Blood miRNA Kit (PreAnalytiX GmbH; Hombrechtikon, Switzerland) following the manufacturer’s instructions. RNA yield was quantified using the Qubit RNA Broad Range Assay Kit (Invitrogen, Thermo Fisher Scientific Inc.; Waltham, MA, USA) and the Qubit 2.0 Fluorometer (Invitrogen, Thermo Fisher Scientific Inc.; Waltham, MA, USA) prior to storage at −80 °C. To generate first strand cDNA, the total RNA samples were reverse transcribed using the SuperScript VILO cDNA Synthesis Kit (Invitrogen, Thermo Fisher Scientific Inc.; Waltham, MA, USA) following the manufacturer’s instructions. TaqMan probe-based quantitative PCRs (qPCRs) were performed to relatively quantify gene expression levels of *GDAP1*. TaqMan Gene Expression Master Mix (Applied Biosystems, Thermo Fisher Scientific Inc.; Waltham, MA, USA) was used in conjunction with TaqMan Gene Expression Assays (Applied Biosystems, Thermo Fisher Scientific Inc.; Waltham, MA, USA), which contain specific primer probes for *GDAP1* (Assay ID: Hs00184079_m1) and *ELOF1* as the internal standard for normalization (Assay ID: Hs00361088_g1). Each qPCR was run in triplicates with a reference sample, and a negative control. The 2^−ΔΔCT^ method was used to quantify *GDAP1* gene expression.

### 4.5. Statistical Analysis

All data analyses were performed using the software environment R (Version 4.3.2., last accessed in December 2024). Statistical tests, which are available within the R package ggpubr [66]. These were used depending on the analysis specified in the following sections.

Gene expression of participants was identified through the 2^−ΔΔCT^ method of relative quantification [67] to determine differences in *GDAP1* expression between patients suffering from AD and healthy individuals. The resulting *GDAP1* expression values (RQ) were subsequently log_2_ transformed to obtain log_2_ fold change (log_2_FC) values used for analysis.

To investigate confounding effects of age, sex, group, and smoking behavior on DNAm in blood and saliva and H3K4me3-FC in patients and controls, a linear mixed effects model was applied using the formula: lmer(variable~Age + Gender + Smoking + Type × Timepoint + (1|Sample)). Data were log-transformed prior to ensure normal distribution.

The normality of data was tested using the Shapiro–Wilk test (Appendix A). Testing with the Shapiro–Wilk method revealed mainly non-normal distributions for all variables. Therefore, the comparison of the trait medians between the independent groups was performed using the Wilcoxon Mann–Whitney rank sum test or *t*-test, as stated in the descriptions of the respective table or figure (Appendix A). A chi-square test was performed to determine the statistical differences between patients and control individuals in regard to smoking and sex (Table 1).

When the Wilcoxon Mann–Whitney rank sum test or *t*-test was used to compare the DNAm levels between (sub)groups, the Benjamini–Hochberg procedure [39] was used to correct for multiple testing. Spearman correlation analysis was used to compare continuous variables. In case of multiple tests Benjamani–Hochberg correction was performed and an ad-justed *p*-value was calculated for the respective number of tests. An adjusted *p*-value (p.adj.) < 0.05 was considered as significant.

Schemes were created using Biorender, Adapted from “The Drug Discovery Process”, by BioRender.com (2024). Retrieved from https://app.biorender.com/biorender-templates (accessed on 23 December 2024).

## Figures and Tables

**Figure 1 ijms-26-01623-f001:**
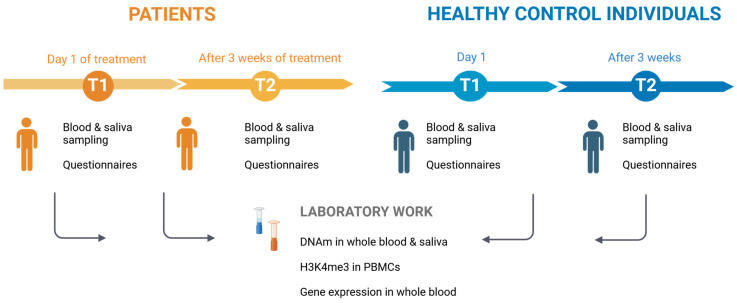
Schematic overview of the study. T1 samples were acquired at day 1 of treatment (AUD patients) or first day of the individual recruitment (Healthy control individuals), respectively. T2 samples were acquired after three-week detoxification therapy (AUD patients) or three weeks after first sampling (Healthy controls individuals). Created with Biorender.com [30].

**Figure 2 ijms-26-01623-f002:**
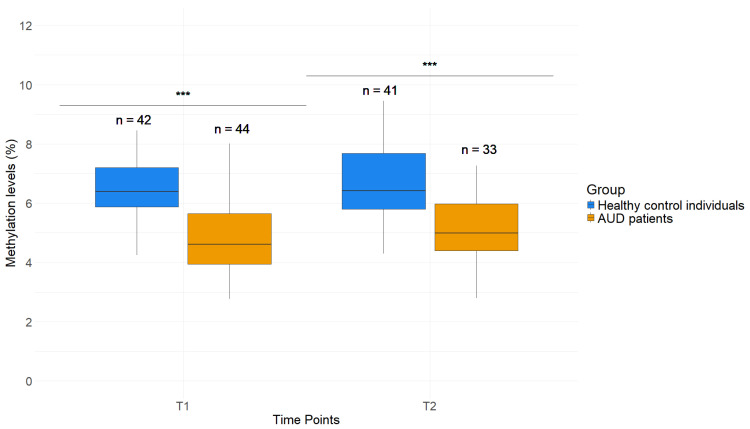
*GDAP1* methylation levels in blood of AUD patients and healthy control individuals. DNAm levels of *GDAP1* in % at T1 and T2. Students *t*-test, *** (*p* < 0.001). *p*-Values were corrected by Benjamini–Hochberg correction.

**Figure 3 ijms-26-01623-f003:**
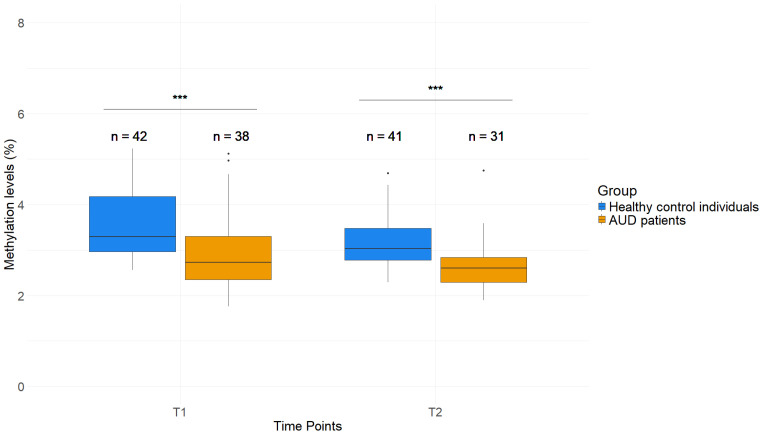
*GDAP1* methylation levels in saliva of AUD patients and healthy control individuals. DNAm levels of *GDAP1* in % at T1 and T2. Wilcoxon rank sum test. *** (*p* < 0.001). *p*-Values were corrected by Benjamini–Hochberg correction.

**Figure 4 ijms-26-01623-f004:**
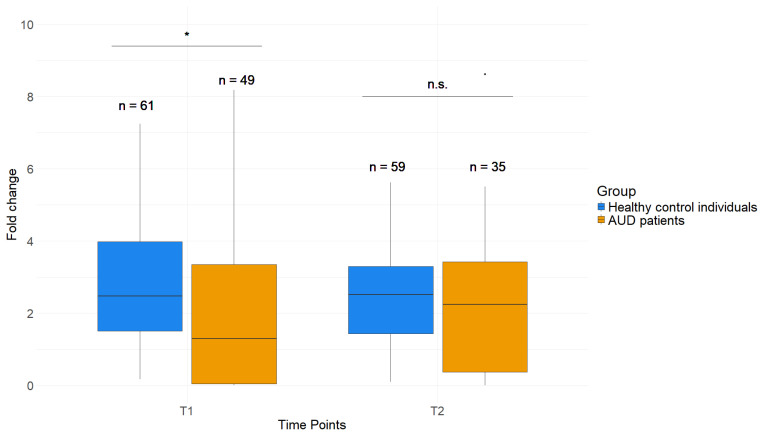
H3K4me3 fold change in AUD patients and healthy control individuals at T1 and T2. Wilcoxon rank sum test. * (*p* < 0.05), n.s. (*p* = not significant). *p*-Values were corrected by Benjamani–Hochberg correction.

**Figure 5 ijms-26-01623-f005:**
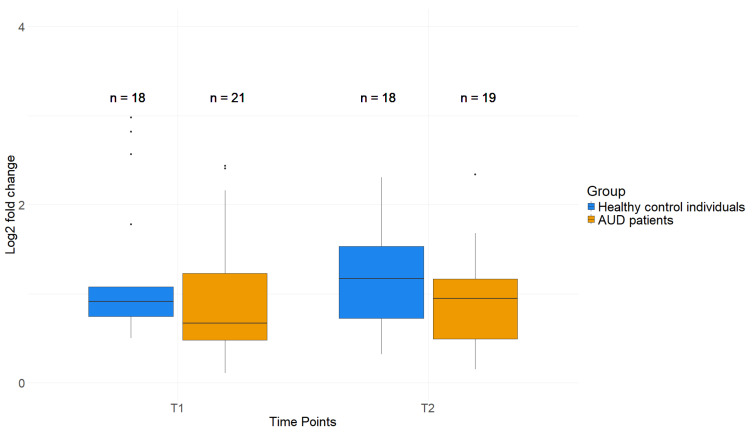
*GDAP1* expression in AUD patients before and after therapy compared to healthy control individuals. Log2 fold change at T1 and T2. Wilcoxon rank sum test.

**Table 1 ijms-26-01623-t001:** Demographics and alcohol related questionnaire scores of patient and healthy control participants per timepoint. Students *t*-test and Wilcoxon rank sum test were used for smoking AUDIT, OCDS, and GSI. Chi-square test was used for sex and age.

		Patients	Healthy Control Individuals	*p*-Values
Sample Size (n)		74	86	
Sex	Female	26 (35.135%)	55 (63.953%)	*p* < 0.001
	Male	48 (64.865%)	31 (36.047%)	*p* < 0.001
Age	All	49.9 ± 11.5	39.1 ± 13.8	*p* < 0.001
	Female	49.4 ± 13.3	37.7 ± 12.7	*p* < 0.001
	Male	50.1 ± 10.5	41.6 ± 15.4	*p* = 0.006
Smokers		58 (79.378%)	4 (4.651%)	*p* < 0.001
AUDIT	T1	27.1 (±7.5)	2.6 (±1.9)	*p* < 0.001
	T2	22.6 (±9.0)	2.9 (±2.0)	*p* < 0.001
OCDS	T1	21.5 (±7.3)	1.7 (±1.8)	*p* < 0.001
	T2	13.4 (±6.3)	1.5 (±1.9)	*p* < 0.001
GSI	T1	65.9 (±9.5)	45.7 (±8.9)	*p* < 0.001
	T2	55.63 (±11.2)	44.5 (±9.1)	*p* < 0.001

**Table 2 ijms-26-01623-t002:** Results of linear mixed model analysis accounting for fixed and random effects. Data were log-transformed to ensure better model fit. n.s. (*p* > 0.05), * (*p* < 0.05), ** (*p* < 0.01), *** (*p* < 0.001).

	a. Blood DNA Methylation	b. Saliva DNA Methylation	c. H3K4me3 Fold Change
Group (patients/healthy controls)	*p* < 0.001 (***)	*p* < 0.001 (***)	*p* = 0.015 (**)
Sex	*p* = 0.05 (*)	n.s.	n.s.
Age	n.s.	*p* = 0.003 (**)	n.s.
Smoking	n.s.	n.s.	n.s.
Timepoint	n.s.	*p* = 0.01 (**)	n.s.
Time × timepoint	n.s.	n.s.	n.s.
Random effect standard deviation	0.1699	0.1124	1.030
Residual standard deviation	0.1297	0.1731	1.315

## Data Availability

The original contributions presented in this study are included in the article/Appendix A. Further inquiries can be directed to the corresponding author(s).

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
