# Peer review of "GDAP1 Is Dysregulated at DNA Methylation and H3K4me3 Levels in Alcohol Use Disorder"

_ijms, 2025, doi:10.3390/ijms26041623_

Round 1
Reviewer 1 Report
Comments and Suggestions for Authors
The manuscript by Kawecka et al. deals with epigenetic disregulation of GDAP1 (encoding ganglioside induced differentiation associated protein 1) in alcohol use disorder (AUD) patients. More precisely, the authors study differences in methylation (DNAm) of 3 CpGs in the promoter, and H3K4me3 level in the 1st exon, of GDAP1 in whole blood, saliva or PMBC samples of AUD patients and healthy control to establish if these epigenetic features can serve as diagnostic biomarkers of AUD.
The Authors report that AUD patients have significantly lower GDAP1 promoter DNAm level in whole blood and saliva, lower H3K4me3 level in PMBC while GDAP1 expression is not statistically different from that in healthy controls. A 3 week abstinence therapy caused an increase in H3K4me3 occupancy and abolished the difference between controls and AUD patients while DNAm and GDAP1 expression remained unchanged.
Major:
-
The scope of DNAm analysis is limited to 3 CpGs previously analyzed. Although the level of methylation of individual CpG sites in a CpG island is usually highly correlated, a larger range of methylation analysis would be of advantage in this type of study.
-
The studied promoter and 1st exon regions of GDAP1 are said to be both near the TSS but the genomic coordinates of these regions (lines 345 and 375) are almost 1 million base pair apart. Certainly, one of these coordinates is false.
-
The H3K4me3 level has been established in PMBC while GDAP1 expression was studied in whole blood (WB). The cell content of both samples is different e.g. neutrophiles present in WB may have different GDAP1 expression than PMBC. Thus, the conclusions regarding 1) the lack of effect of H3K4me3 level on expression and 2) the apparent paradox between DNA hypomethylation and decreased H3K4me3 (line 224-227) may be invalid. At least GDAP1 expression in PMBC should be studied to validate this issue.
-
The ChIP analysis should also contain a negative control that is a sample immunoprecipitated with rabbit IgG.
Minor
- The difference between DNAm or H3K4me3 in AUD patients and healthy controls, although statistically significant between groups, is very small. Please, comment how such parameters could help in AUD diagnosis of an individual person?
- A scheme of the study should be briefly summarized at the end of Introduction or the begining of Results, otherwise terms such as „both time points”, T1, T2 etc. are not clear without referring to Methods.
- The term „whole blood DNAm” without reference to GADP1 promoter as, for example in line 121, suggests whole genome DNA methylation analysis.
-was the control group also sujected to abstinence therapy? See legend to fig 4.
- Minor editing or English errors should be corrected (e.g. line 24 -word missing)
Author Response
Thank you for the thorough review! Please find our reply attached below.
Kind regards,
Emilia Kawecka

Reviewer 2 Report
Comments and Suggestions for Authors
Manuscript ID: ijms-3423459-peer-review-v1
The Authors of the manuscript entitled: “GDAP1 is dysregulated at DNA methylation and H3K4me3 levels in alcohol use disorder” aimed to explore the epigenetic dysregulation of GDAP1 in alcohol use disorder (AUD) patients by investigating DNA methylation (DNAm) of the promoter region in whole blood and saliva, as well as H3K4me3 in peripheral blood mononuclear cells (PBMCs). In addition, the impact of those modifications on gene expression levels have been analyzed and the effects of a three-week inpatient detoxification treatment have been evaluated.
The study was conducted on 74 AUD patients and 86 healthy controls. AUD patients were subjected to a 3-week inpatient treatment program, where they went through alcohol withdrawal and therapy. Blood and saliva samples as well as questionnaires were collected before (T1) and after the 3-week therapy (T2). The analyzes performed in the study included: GDAP1 DNA-methylation determination in whole blood and saliva, H3K4-trimethylation analysis in GDAP1 in PBMCs, and quantification of gene expression levels of GDAP1.
The main findings of the study are: 1. the GDAP1 DNAm level was significantly lower in AUD patients in comparison to healthy control individuals in whole blood samples, 2. GDAP1 promoters display lower DNAm levels in saliva of AUD patients compared to healthy control individuals, and 3. reduced H3K4me3 levels near TSS of AUD patient PBMCs compared to healthy control individuals. In addition it was found that the 3-week treatment influenced fold change of the histone modification in a positive way, reverting its values to control-comparable levels. GDAP1 DNAm did not show this reversion in both blood and saliva of AUD patients. The gene expression levels of GDAP1 in whole blood of AUD patients was not changed as compared to controls.
The manuscript presents valuable study and interesting results. Some minor issues should be addressed before the publication:
- A short explanation of what and when was analyzed in the study should be presented at the beginning of results section for clarity.
- The abbreviations that show up in the results section, e.g. AUDIT, OCDS, GSI, should be explained the first time they appear in the text.
- Figure 1, 2 and 3 – legends of the figures should contain information what values are presented on the graphs. Figure 2 – “%” sign is missing from y axis description.
- Regarding reference 27 – please provide references to publications documenting the association of GDAP1 with Charcot-Marie-Toorth Disease
- The language of the manuscript should be improved. Some typing mistakes are noticeable, e.g. missing space (line 59), additional dot (line 37), glutathione-S-tranferase should replace glutathione-s-tranferase (line 77), rewrite “catalyzes the bond” (enzyme can catalyze a reaction not a bond) (line 77). Consider the change of “epigenetically dysregulated marks in GDAP1” to “dysregulated epigenetic marks in GDAP1” (line 202), and “hypomethylation of saliva” to “hypomethylation in saliva” (line 203.
Author Response
Thank you for your thourough review! Please find our reply attached below.
Kind regards,
Emilia Kawecka

Round 2
Reviewer 1 Report
Comments and Suggestions for Authors
All discrepancies and formal errors have been corrected and the suggested additions to the text have been implemented. Nonetheless, the main flaws of the manuscript i.e. 1. a limited region of DNAm analysis and 2. different material on which H3K4me3 level and GADP1 expression were assessed, remained.
Author Response
Dear reviewer 1,
We are thankful for your review. Please find our answer attached below.
Best regards,
Emilia Kawecka

Round 3
Reviewer 1 Report
Comments and Suggestions for Authors
Although the Authors agree with my opinion that due to different biological material used in some experiments the conclusions are of limited value, they seem to be determined to publish their results while making the potential readers aware of the concerns and reservations as to their interpretaton. The provided explanations seem too long and introduce additional flaws, see below:
Line 283:The limitation of comparing GDAP1 DNAm and gene expression in whole blood to H3K4me3 in
PBMCs makes it harder to form causal relationships between two different tissues the two epigenetic features and gene expression.
Line 287: Therefore, it is of utmost necessity to assess GDAP1 DNAm and gene expression in PBMCs in future studies to confidently say that the aberrant pattern of both reduced H3K4me3 occupancy and DNAm in GDAP1, combined with seemingly, although not statistically significant, lower gene expression is present in AUD patients and is not attributable to tissue specific differences.
Please check the spelling of promoter/promotor
Author Response
Dear reviewer 1,
Thank you for your positive review. Please find our answer and implementations below.
Best regards,
Emilia Kawecka
